# Novel In Vitro Assay of the Effects of Kampo Medicines against Intra/Extracellular Advanced Glycation End-Products in Oral, Esophageal, and Gastric Epithelial Cells

**DOI:** 10.3390/metabo13070878

**Published:** 2023-07-24

**Authors:** Takanobu Takata, Yoshiharu Motoo

**Affiliations:** 1Division of Molecular and Genetic Biology, Department of Life Science, Medical Research Institute, Kanazawa Medical University, Uchinada 920-0293, Ishikawa, Japan; 2Department of Medical Oncology and Kampo Medicines, Komatsu Sophia Hospital, Komatsu 923-0861, Ishikawa, Japan; mot@sophia-hosp.jp

**Keywords:** Kampo medicines, digestion, absorption, metabolism, oral epithelial cells, esophageal epithelial cells, gastric epithelial cells, advanced glycation end-products, in vitro

## Abstract

Kampo medicines are Japanese traditional medicines developed from Chinese traditional medicines. The action mechanisms of the numerous known compounds have been studied for approximately 100 years; however, many remain unclear. While components are normally affected through digestion, absorption, and metabolism, in vitro oral, esophageal, and gastric epithelial cell models avoid these influences and, thus, represent superior assay systems for Kampo medicines. We focused on two areas of the strong performance of this assay system: intracellular and extracellular advanced glycation end-products (AGEs). AGEs are generated from glucose, fructose, and their metabolites, and promote lifestyle-related diseases such as diabetes and cancer. While current technology cannot analyze whole intracellular AGEs in cells in some organs, some AGEs can be generated for 1–2 days, and the turnover time of oral and gastric epithelial cells is 7–14 days. Therefore, we hypothesized that we could detect these rapidly generated intracellular AGEs in such cells. Extracellular AEGs (e.g., dietary or in the saliva) bind to the receptor for AGEs (RAGE) and the toll-like receptor 4 (TLR4) on the surface of the epithelial cells and can induce cytotoxicity such as inflammation. The analysis of Kampo medicine effects against intra/extracellular AGEs in vitro is a novel model.

## 1. Introduction

Kampo medicines, the Japanese traditional medicines, developed based on Chinese traditional medicines. The major differences are that Japanese Kampo medicine is integrated with Western (modern) medicine and used in a unified medical system, the amount of crude drugs used in prescriptions is small, and the quality of medical extract preparations is extremely high. It is high, not ideological, but practical like “Housoushoutai (Japanese original treatment of Kampo medicine)” [1]. Although up to the 19th century, doctors and individuals used to extract and apply components from crude drugs as needed, modern Kampo medicines are produced from extracts following manufacturing methods that are governed by a number of national laws in Japan [2,3]. Officially recognized current Kampo medicines are stipulated in the Japanese Pharmacopoeia, and their quality must comply with legal provisions [2,3]. Modern medical methods have proven Kampo preparations are beneficial in various clinical fields, and their value is being reassessed following their successful use in cancer support care [4]. Over 500 randomized controlled trials using Kampo preparations have been published as structured abstracts and clinical evidence for their efficacy is accumulating [5,6]. The slogan “Kampo for cancer supportive care” is supported by industry, academia, and the Japanese government [7]. Kampo preparations are characterized by containing multiple ingredients and should be administered orally. They often contain crude drugs with different compound contents [2]. Analysis using three-dimensional high-performance liquid chromatography (3D-HPLC) of compounds in Kampo medicines such as Goshajinkigan and Ninjin’yoeito have indicated a wide range of contents [8,9,10]. Although the main and most common elements contained in the principal components of some Kampo medicines have been identified [11], complete component identification has so far not been achieved. Because Kampo medicines are usually administered orally, the components should be digested in the stomach, absorbed in the small intestine, and metabolized in the liver and other organs [12]. The metabolized compounds have wide-ranging effects throughout the body [13]. Although the effects of Kampo medicines have been investigated in animal models with some success [14,15], the precise mechanisms involved have so remained elusive. Furthermore, in vitro assays of Kampo medicines are more difficult than in vivo assays because such laboratory experiments cannot account for normal digestion, absorption, and metabolism processes [16]. This makes in vitro assays using human oral administration conditions an attractive novel assay system. Oral, esophageal, and gastric epithelial cells have so far been used in this type of assay [17,18,19].

We investigated reports of the use of these three kinds of epithelial cells in assays of extracts of Kampo medicines and other natural products. To reproduce the absorption step, small intestinal cells should be selected for such assays, but the factor of acid hydrolysis of Kampo medicines may make this difficult. Furthermore, we focused on the strong performance of this assay system against intracellular and extracellular advanced glycation end-products (AGEs). AGEs are generated from glucose, fructose, and their metabolites, and promote lifestyle-related diseases such as diabetes and cancer [20,21,22,23]. Whole intracellular AGEs in cells cannot be analyzed with current technology, although some have been detected in the liver [24], brain [24], kidney [24], lung [25], heart [26,27,28], skeletal muscle [29], skin [30], and pancreatic islets [31]. Some AGEs such as glyceraldehyde-derived pyridinium (GLAP), argpyrimidine, and *N*^δ^-(5-hydro-5-methyl-4-imizazolon-2yl)-ornithine (MG-H1) can be generated for a duration of 1–2 days [32,33,34], and the turnover time of oral and gastric epithelial cells is 7–14 days [35,36]. Therefore, we hypothesize that it should be possible to detect rapidly generated intracellular AGEs in these cells. The receptor for AGEs (RAGE) [17,18,37,38,39] and the toll-like receptor 4 (TLR4) [40,41,42] are expressed on the surface of the epithelial cells of interest. Extracellular AGEs (e.g., dietary [20,21] or in saliva and blood [43,44]) bind to RAGE and TLR4 and can cause cytotoxicity, such as inflammation [37,38,39,40,41,42]. Therefore, an in vitro assay using oral, esophageal, and gastric epithelial cells could assess the effects of samples such as Kampo medicines against the AGEs-RAGE/TLR4 combination. We hypothesize that the action mechanisms of Kampo medicines can be studied against intra- and extracellular AGE using this novel in vitro assay.

## 2. Components in Kampo Medicines and Their Digestion, Absorption, and Metabolism

Because Kampo medicines contain various crude drugs, each with a number of different components [2], research has been undertaken to characterize the main components and their cell- and organ-level effects. Kishida et al. and Nakanishi et al. used 3D-HPLC to analyze the components of Gosyajinkigan extract, and reported moconiside, (+)-catechin, loganin, paconiflorin, penta-*O*-gallolyglucose, benzozylmesaconie, cinnamic acid, isoacteoside, benzyoyl lpaeconiflorin, cinnamaldehyde, 16-ketoalisol A, and paenol. Further as yet uncharacterized compounds were also detected [8,9]. Hosogi et al. reported that paeoniflorin, hesperidin, and glycyrrhizic acid were effective chemical markers in analyzing the components in Ninjin’yoeito extract; they showed the presence of a number of compounds, many of which were not identified [10]. Jin et al. reported the main components in 34 Kampo medicines and their major elements, but were unable to elucidate the complete mechanisms of the effects of each medicine [11]. Each component of Kampo medicines is digested in the stomach, absorbed in the small intestine, metabolized in the liver, and passes into the blood, which induces various effects on organs [12,13] (Figure 1). Components in Kampo medicines may be affected in the mouth through the activity of enzymes such as α-amylase [45], and this enzyme reaction may be the first step in their respective metabolism. The mechanisms of Kampo medicine effects have, therefore, frequently been investigated, with in vivo animal models treated with oral administration of the extract [8,9,10,14,15].

## 3. Novel In Vitro Assay of Kampo Medicines Effects

### 3.1. Effects of Kampo Medicine Extracts and Other Natural Products on Dermal, Adipose, and Cardiac Cells In Vitro

We previously investigated the effects of the extract of Saikokeishito on the premature senescence of human dermal fibroblasts [16]. In that study, the activity of p53 decreased and that of adenosine monophosphate (AMP)-activated protein kinase (AMK) α1 (AMPKα1) increased in the treatment of hydroperoxide-induced premature senescence. Because p53 promotes senescence and while AMPKα1 inhibits it, Saikokeishito may act to inhibit cellular senescence. Matsuda et al. reported that an extract of a kava (*Piper methysticum)* rhizome, a traditional medicine of Micronesia, induced the generation of melanin in B16F1 (murine melanoma cell line) [46]. Because melanocytes generate and secrete melanin, the authors measured intracellular melanin in cell lysates and extracellular melanin in the supernatant when cells were incubated. Yamagishi et al. used a DNA microarray system and a transcriptase-polymerase chain reaction (RT-PCR) assay to analyze RNA expression in rat white adipocytes that were treated with the extracts of Orengedokuto, Bofutsushosan, and Boiogito [47]. Poindexter et al. analyzed the effects of *Panax ginseng* (a crude drug used in Kampo medicines [2]) on intracellular calcium levels and the beating of rat primary cardiomyocytes [48].

Although these four investigations demonstrated the effects of Kampo medicines or other traditional medicines on cells in vitro, the factors of digestion, absorption, and metabolism were not incorporated in the assessments [12,13]. The identified components may, thus, induce different effects on cells in vivo, which is a limitation of these results.

### 3.2. Effects of Kampo Medicine Extracts on Oral Epithelial Cells In Vitro

Miyano et al. investigated the effects of Hangeshashinto extract on primary human oral keratinocyte [49]. They found that this extract enhanced scratch-induced keratinocyte migration through mitogen protein kinase (MAPK) and C-X-C chemokine receptor 4 (CXCR4). Furthermore, it upregulated C-X-C motif chemokine ligands 12 (CXCL12) through extracellular-signal-regulated kinase (ERK). Using in vitro examination, they were able to employ scratch keratinocyte cells as a model of the lack of oral epithelial area and easily assess the confluence in the wound area. Hsu et al. investigated the effect of San-Zhong-Kui-Jian-Tang (Japanese name: Sanshukaigento) on SAS, OC3, and OEC-M1, which belong to oral squamous cell lines [50]. They showed a substantial increase in the proportion of cells arrested in the S phase, whereas the migration of SAS and OC3 cells was inhibited and their proportions in the G2/M phase decreased.

### 3.3. Effects of Natural Product Extracts on Esophageal Epithelial Cells In Vitro

Krestry et al. evaluated how the esophageal epithelial cell lines OE33 and JHAD1 responded to exposure to an acidified bile cocktail (pH = 4). They reported that cell types were acid-sensitive and responded with rapid cell death (40% and 50%, respectively) within 24 h post-treatment. When treating JHAD1 acid-resistant cells (JHAD1-AR) with cranberry-derived proanthocyanin extract (C-PAC), they found that this significantly induced cell death via late apoptosis and substantial cellular necrosis [51]. Tesfaye et al. researched the cytotoxicity of 80% methanol extracts of 22 plants against various cell lines and selected 4 extracts that evaluated IC_50_ against KYSE-70, an esophageal epithelial cell [52].

### 3.4. Effects of Natural Product Extracts on Gastric Epithelial Cells In Vitro

Matsuhashi et al. investigated a novel rice extract (Rice Power Extract No. 101) using gastric epithelial cells treated with ethanol (ulcer model) in vivo and in vitro, showing that this reduced the ulcer index. In contrast, 3% ethanol decreased the cell viability of RGM-1 cells (rat gastric epithelial cell line), and rice extract was shown to be able to recover this cell viability. Furthermore, rice extract significantly abolished the suppressive effects of the 3% ethanol against RGM-1 cells in a wound restoration assay [53]. Wang et al. investigated the effects of Hekikaso (*Trichodesma khasianum* Clarke) leaves on ethanol-induced gastric mucosal injury in vivo and in vitro by preparing an 80% ethanol extract of Heikaso leaves (80EETC) [54]. 80EETC was found to moderate gastric injury induced by oral administration of 100% ethanol in BALB mice, but promoted the migration of RGM-1 cells in a wound-healing assay without, at 0%, and with 5% ethanol.

Both of these experiments suggest that a strength of this assay is in the capacity for comparing in vitro with in vivo experimental outcomes.

### 3.5. Application of the Novel In Vitro Assay Using Oral, Esophageal, and Gastric Epithelial Cells to Assess the Effects of Kampo Medicines

In vitro assay systems using oral, esophageal, and gastric epithelial cells appear suitable for investigating the direct effects of Kampo medicines. In this approach, the digestion, absorption, and metabolism steps can be removed (Figure 2), which is a benefit with respect to the investigation of Kampo medicines, and the results of both in vitro and in vivo examinations can be compared [53,54]. However, this assay system is not complete because (i) oral and gastric epithelial cells secrete α-amylase and hydrochloric acid [45,55,56] and (ii) under normal circumstances, various components in Kampo medicines are digested, absorbed, and metabolized in specific organs, and their metabolized compounds can affect the epithelial cells of interest. Despite these limitations, we consider that this in vitro assay system is a valuable novel approach for the investigation of the effects of Kampo medicines. If samples of Kampo medicines treated with hydrochloride acid can be prepared [55,56], then the effects of such medicines as digested in the stomach can be examined (Figure 2).

## 4. Use of the Novel In Vitro Assay to Assess the Effects of Kampo Medicines against Intracellular AGEs

### 4.1. Various Intracellular AGEs

During the early investigation of AGEs, six categories were established: glucose-derived AGEs (Glc-AGEs, AGE-1), glyceraldehyde-derived AGEs (GA-AGEs, AGE-2), glycolaldehyde-derived AGEs (AGE-3), methylglyoxal-derived AGEs (MGO-AGEs, AGE-4), glyoxal-derived AGEs (GO-AGEs, AGE-5), and 3-deoxyglucosone-derived AGEs (3DG-AGEs, AGE-6) [20]. These groups were named based on the saccharide/metabolite/intermediate that was the origin of the AGE’s structure. Subsequent research has altered the recognition of AGEs. It was reported that GLAP [57,58], MG-H1 [59], and argpyrimidine [60] were generated from glyceraldehyde in test tubes (categorized as GA-AGEs), although MG-H1 and argpyrimidine had been categorized as MGO-AGEs. Senavirathna et al. identified and quantified GLAP, MG-H1, and argpyrimidine-modified proteins in a human pancreatic ductal cell line (PANC-1) and human normal pancreatic ductal epithelial (HPDE) cells treated with glyceraldehyde [32]. In contrast, Wang et al. reported that *N*^ε^-(carboxymethyl)-lysine (CML)-modified proteins in H9c2 (a rat cardiomyocyte cell line), which were treated with methylglyoxal, increased in Western blot analysis, but did not categorize them as MGO-AGEs [61]. Basakal et al. used gas chromatography–mass spectroscopy (GC-MS) to show that both CML and *N*^ε^-(carboxyethyl)-lysine (CEL) were generated from methylglyoxal in the test tube [62]. They investigated various generation conditions, such as the concentration of lysine, reaction time, and temperature, and found that CEL was generated at much higher proportions than CML under the same condition (CEL >> CML). Litwinowicz et al. determined the structure of novel melibiose-derived AGEs (MAGE) and classified them as AGE-10 [63].

Due to the increased availability of technology suitable for the identification and quantification of AGEs, such as nuclear magnetic resonance [64,65], Western blot [29,61], immunostaining [25,31], slot blot [66,67,68], ELISA [27,69,70,71], GC-MS [62], MALDI-MS [34,72], and ESI-MS [32,73,74,75], individual AGEs can be increasingly investigated to reveal novel findings.

### 4.2. Use of the Assay to Assess the Effects of Kampo Medicines against Rapidly Generated Intracellular AGEs

Current technology does not allow the analysis of complete intracellular AGEs in cells; however, more limited analysis of AGEs is possible under specific conditions. We focused on several AGEs that can be generated to persist for a short period (1–2 days) [20,32,74,76]; in comparison, the turn-over time of oral and gastric epithelial cells is 7–14 days [35,36]. Methylglyoxal, glyceraldehyde, and glycolaldehyde are able to rapidly generate AGEs [59,60,74]. Senavirathana et al. identified GLAP-, MG-H1-, and argpyrimidine-modified proteins in both PANC-1 and HPDE cells treated with glyceraldehyde for 48 h [32]. Nokin et al. and Oya-Ito et al. prepared argpyrimidine-, MG-H1-, and CEL-modified recombinant human heat shock protein (HSP) 90 and HSP27 at 37 °C for 24 h in a test tube [33,34]. Suzuki et al. reported that glycolaldehyde was able to generate CML in MC3T3-E1 (murine osteoblast cell line) within 24 h and that CML may be rapidly generated if cells quickly produce a high amount of glycolaldehyde [74]. Kinoshita et al. prepared Glc-AGEs-modified BSA at 37 °C for 4 weeks in a test tube [28]. In the investigation of intracellular AGEs in oral and gastric epithelial cells, rapidly generated AGEs (e.g., GLAP-, MG-H1, argpyrimidine-, and CEL-modified proteins) may be suitable as targets. Oya-Ito et al. reported that some arpyrimidine-modified proteins were generated in RGM-1 and RGK-1 (an N-methyl-N’-nitro-N-nitrosoguanidine-induced mutant of a tumor in RGM-1) cells incubated in normal Dulbecco’s modified Eagle medium and Ham’s F-12 medium (DMEM/F12) [34]. Because they did not treat the cells with methylglyoxal or glyceraldehyde, these argpyrimidine-modified proteins may have been generated from intracellular methylglyoxal/glyceraldehyde. Based on these reports, it may be possible to investigate the relationship between hyperglycemia, ulcers, and rapidly generated intracellular AGEs. Furthermore, the action mechanisms of Kampo medicines against oral and gastric epithelial cells that rapidly generate intracellular AGEs can be revealed (Figure 3).

## 5. Use of the Novel In Vitro Assay to Assess the Effects of Kampo Medicines against Extracellular AGEs

### 5.1. Extracellular AGEs Released from Organs and Ingested from Foods and Beverages

Intracellular AGEs in some organs that are released into extracellular fluids, such as saliva, blood, and urine [43,44,77,78], are termed extracellular AGEs. The relationship between extracellular AGEs and diseases, such as diabetes and atherosclerosis, has been investigated [43,44,78]. If these relationships can be proved, extracellular AGEs could represent a biomarker against disease.

Litwinowicz et al. showed that a MAGE they synthesized, identified, and named AGE-10 existed in blood, and proved that it was associated with alcoholic hepatitis [65]. Dietary AGEs have received attention as a type of extracellular AGEs that are generated in foods and beverages and ingested daily [20,21,39,79,80]. CML is the product most commonly used to estimate dietary AGEs in the body and has a well-characterized structure [43]. Although CML may be generated in the body only slowly [18], there is the possibility that humans take in a large amount of CML as dietary AGEs. Chen et al. focused on CML, CEL, and MG-H1 as dietary AGEs consumed by humans [21]. Extracellular AGEs can induce cytotoxicity, such as inflammation, via RAGE [81,82] and TLR4 receptors [83,84]. While studies have investigated the interaction of dietary AGEs, such as CML-modified BSA, with the cells of some organs (e.g., liver, heart, kidney, and brain), these experiments lacked the three steps of digestion, absorption, and metabolism.

### 5.2. Using the Novel In Vitro Assay with Oral, Esophageal, and Gastric Epithelial Cells to Assess the Effects of Kampo Medicines against Extracellular AGEs

RAGE is expressed on the surface of oral [17], esophageal [17], and gastric [41] epithelial cells, allowing extracellular AGEs to bind to and induce effects against these cells. TLR4 is also expressed in all three cell types [38,39,40,41,44,85]. In the investigation of the effects of extracellular AGEs via RAGE/TLR4 in these cells in vitro, digestion, absorption, and metabolization can be excluded, as discussed above, and an assessment of the effects of samples such as Kampo medicines against AGEs-RAGE/TLR4 combinations is, thus, practical (Figure 4). We expect that the investigation of dietary AGEs and Kampo medicines will be promoted using this assay.

### 5.3. Potential of the Novel Assay

This novel assay is suitable for the investigation of the relationships between lifestyle-related diseases or cellular dysfunctions, intracellular/extracellular AGEs, and the effects that Kampo medicines may exert against them. The effects of Kampo medicines can be assessed while bypassing the influence of digestion, absorption, and metabolism on the examined compounds. While other, unrelated natural products, such as different Eastern traditional medicines, may also be assessed in this manner, we consider that Kampo medicines represent more suitable candidates due to their ensured quality and conformance to the Japanese Pharmacopoeia [2,3]. We suggest that the mechanisms of intracellular/extracellular AGEs have the potential to induce ulcers and to promote tumors in three kinds of epithelial cells. Although the cytotoxicity of intracellular AGEs in oral and esophageal cells remains unclear, clear relationships between such AGEs and gastric ulcers and tumors have been reported [34,86,87]. Naito et al. and Takagi et al. reported that an argpyrimidine-modified peroxidoxin VI protein was detected in gastric cells in diabetic model mice, and found that methylglyoxal modification of proteins delayed gastric ulcer healing [86,87]. Oya-Ito et al. detected argpyrimidine-modified HSP25 in RGK-1 cells, but not in RGM-1 cells [34]. Because these AGE-modified proteins are of the argpyrimidine-modified type, we propose that rapidly generated AGEs may be suitable targets for such investigations into AGE associations with lifestyle-related diseases or organ dysfunction. Because this in vitro assay constitutes a simple system, the cells of which can be incubated in media in dishes or plates, researchers can make use of it to treat the cells with metabolites/intermediate compounds from glucose and fructose metabolism and collect both cells and supernatant. Intracellular AGEs can then be identified and quantified by using Western blot [29,61], immunostaining [25,31], slot blot [66,67,68], ELISA [27,69,70,71], GC-MS [62], MALDI-MS [34,72], and ESI-MS [32,73,74,75]. mRNA levels, intracellular protein levels, secreted/released proteins (e.g., cytokine), and the type of cell death (e.g., necrosis or apoptosis) can be determined with PCR [88,89], Western blot [90,91], slot blot [66], ELISA [92,93,94], MALDI/ESI-MS [90], microscope [95,96], or flow cytometry [95,97,98,99]. Specific investigation methods such as the measurement of membrane potential, DNA biosynthesis assessment, cell viability assay, and enzyme assay in microsomes can also be applied because researchers can obtain each reagent and instrument required for examination [100,101,102,103]. Extracellular AGEs may be able to promote ulcers in three kinds of epithelial cells via the AGEs-RAGE/TLR4 axis, although this may also be caused by other factors (such as anti-cancer drug side effects [7]). Researchers may employ AGEs obtained from manufacturers or prepared by themselves as models of AGEs in saliva, blood, food, and beverage against cells [104,105,106], and these methods are applicable to extracellular as well as intracellular AGEs.

## 6. Conclusions

The analysis of the effects of Kampo medicines on intra/extracellular AGEs in three kinds of epithelial cells in vitro is a novel model.

## Figures and Tables

**Figure 1 metabolites-13-00878-f001:**
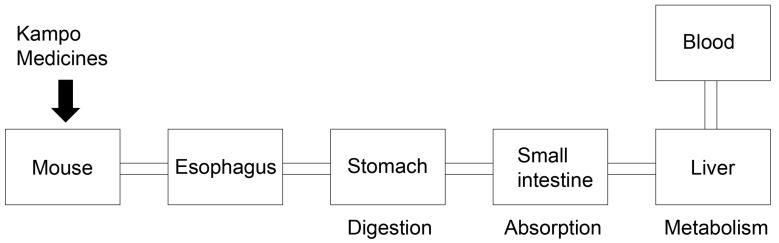
Schematic diagram of progression of Kampo medicines used as oral administration in a mouse model. Medicines are digested in the stomach, absorbed in the small intestine, metabolized in the liver, and passed into the blood.

**Figure 2 metabolites-13-00878-f002:**
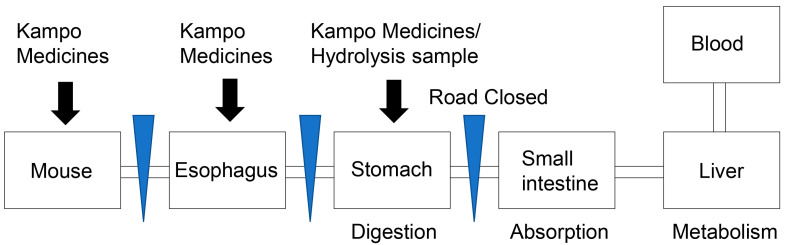
Schematic diagram of the novel in vitro assay for analysis of the effects of Kampo medicines on the complete organism (mouse), esophagus, and/or stomach. In this assay, the steps of absorption and metabolism can be removed, and preparation of hydrolyzed samples allows assessment of stomach-digested medicine. Blue triangles indicate medicine or digestate transportation pathways that can be closed.

**Figure 3 metabolites-13-00878-f003:**
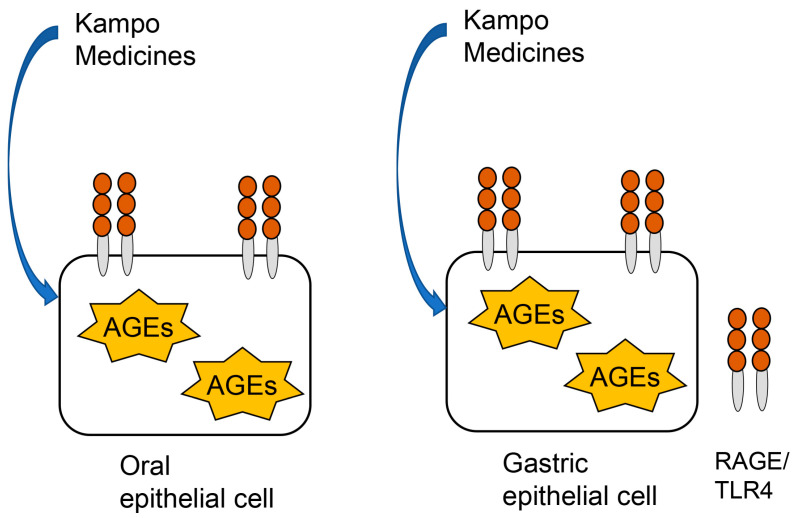
Schematic diagram of application of the novel in vitro assay for assessing the effects of Kampo medicines on intracellular advanced glycation end-products (AGEs) rapidly generated in oral and gastric epithelial cells. RAGE: receptor for AEGs. TLR4: toll-like receptor 4.

**Figure 4 metabolites-13-00878-f004:**
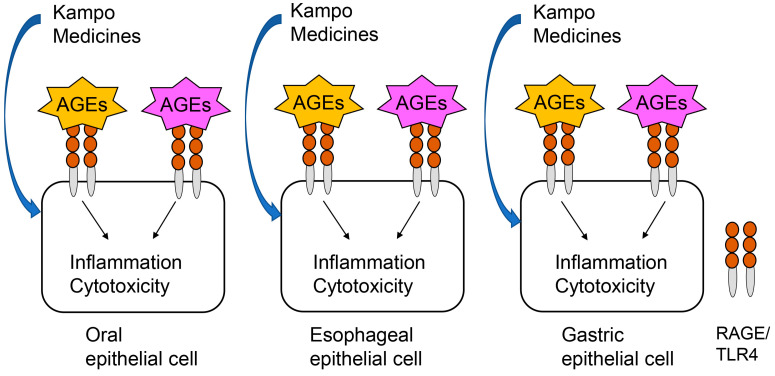
Schematic diagram of application of the novel in vitro assay for assessing the effects of Kampo medicines against extracellular advanced glycation end-products (AGEs) that can bind to receptor for AEGs (RAGE) and/or toll-like receptor 4 (TLR4) in oral, esophageal, and gastric epithelial cells. Yellow AGEs: AGEs released from organs. Pink AGEs: dietary AGEs.

## Data Availability

The data are contained within the article.

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
