# Peer review of "Novel In Vitro Assay of the Effects of Kampo Medicines against Intra/Extracellular Advanced Glycation End-Products in Oral, Esophageal, and Gastric Epithelial Cells"

_metabolites, 2023, doi:10.3390/metabo13070878_

Round 1
Reviewer 1 Report
The authors proposed a new system for in vitro study of Kampo components against intra/extracellular advanced glycation end products in epithelial cells of the oral cavity, esophagus and stomach. In general, the study is interesting and can be claimed by a wide range of researchers. Meanwhile, the authors in the introduction should describe more clearly and in detail the main unique differences of Kampo from the components used in traditional medicine in other eastern regions. Can the proposed system be used to study them? The conclusions seem too laconic, they should reflect not only the quintessence of the results achieved in the article, but also the prospects for their further use and development of their potential in the future. Is it possible to disseminate the developed approaches for conducting and creating kits for analysis using flow cytometry or ELISA analysis?
Author Response
Response Letter to Reviewers’ Comments
Responses to Reviewer 1
Dear Reviewer 1,
Thank you for giving us the opportunity to submit a revised draft of our manuscript titled “Novel in vitro Assay of the Effects of Kampo Medicines against Intra/Extracellular Advanced Glycation End-Products in Oral, Esophageal, and Gastric Epithelial Cells” to the journal Metabolites (manuscript ID: 2506715). We appreciate the time and effort that the reviewers have dedicated to providing their valuable feedback on our manuscript. We are grateful to the reviewers for their thoughtful suggestions and insightful comments on our paper, which have enriched the manuscript and produced a better and more balanced account of the research.
We have inserted the new reference 2, 3, 66-69, 72, 88-108 in the revised manuscript.
Comment 1: In general, the study is interesting and can be claimed by a wide range of researchers. Meanwhile, the authors in the introduction should describe more clearly and in detail the main unique differences of Kampo from the components used in traditional medicine in other eastern regions.
Response 1: We are unable to detail the differences in components between Kampo medicines and other traditional medicines in use in other Eastern regions. However, we focused on Kampo medicines in this article because (i) they have seen long-standing use as Japanese traditional medicines, (ii) the extracts used in formulations are produced by the pharmaceutical industry, (iii) they are recognized as medicines by the Japanese government, and (iv) they are codified in the Japanese Pharmacopoeia, and their quality is required to comply with legal provisions.
We have added the following passage to the Introduction section of the revised manuscript:
“Although up to the 19th century, doctors and individuals used to extract and apply components from crude drugs as needed, modern Kampo medicines are produced from extracts following manufacturing methods that are governed by a number of national laws in Japan [2,3]. Officially recognized current Kampo medicines are stipulated in the Japanese Pharmacopoeia, and their quality must comply with legal provisions [2,3].”.
(Lines 36-41)
We also inserted the new references 2 and 3 in the Introduction section.
Comment 2: Can the proposed system be used to study them?
Response 2: We propose that this novel assay system may also be able to be used for the assessment of other traditional Eastern medicines. However, we consider that Kampo medicines are more suitable for this assay system for a variety of reasons. We inserted this information in the newly added section 5.3 “Potential of the novel assay”.
(Lines 288-291)
Comment 3: The conclusions seem too laconic, they should reflect not only the quintessence of the results achieved in the article, but also the prospects for their further use and development of their potential in the future. Is it possible to disseminate the developed approaches for conducting and creating kits for analysis using flow cytometry or ELISA analysis?
Response 3: Following your suggestions, we have added the following sentences in Section 5. 3.“Potential of the novel assay”.
“This novel assay is suitable for the investigation of the relationships between lifestyle-related diseases or cellular dysfunctions, intracellular/extracellular AGEs, and the effects that Kampo medicines may exert against them. The effects of Kampo medicines can be assessed while bypassing the influence of digestion, absorption, and metabolism on the examined compounds. While other, unrelated natural products such as different Eastern traditional medicines may also be assessed in this manner, we consider that Kampo medicines represent the more suitable candidates due to their ensured quality and conformance to the Japanese Pharmacopoeia [2,3]. We suggest that the mechanisms of intracellular/extracellular AGEs have the potential to induce ulcers and to promote tumors in three kinds of epithelial cells. Although the cytotoxicity of intracellular AGEs in oral and esophageal cells remains unclear, clear relationships between such AGEs and gastric ulcers and tumors have been reported [35,88,89]. Naito et al. and Takagi et al. reported that argpyrimidine-modified peroxidoxin VI protein was detected in gastric cells in a diabetic model mice, and found that methylglyoxal modification of proteins delayed gastric ulcer healing [88,89]. Oya-Ito et al. detected argpyrimidine-modified HSP25 in RGK-1 cells, but not in RGM-1 cells [35]. Because these AGEs-modified proteins are of the argpyrimidine-modified type, we propose that rapidly generated AGEs may be suitable targets for such investigations into AGEs associations with lifestyle-related disease or organ dysfunction. Because this in vitro assay constitutes a simple system, the cells of which can be incubated in medium in dishes or plates, researchers can make use of it to treat the cells with metabolites/intermediate compounds from the glucose and fructose metabolism, and collect both cells and supernatant. Intracellular AGEs can then be identified and quantified by western blot [30,63], immunostaining [26,32], slot blot [68-70], ELISA [28,71-73], GC-MS [64], MALDI-MS [35,74], and ESI-MS [33,75-77]. mRNA levels, intracellular protein levels, secreted/released proteins (e.g. cytokine), and type of cell death (e.g. necrosis, apoptosis) can be determined with PCR [90,91], western blot [92,93], slot blot [68], ELISA [94-96], MALDI/ESI-MS [92], microscope [97,98], or flow cytometry [97, 99-101]. Specific investigation methods such as the measurement of membrane potential, DNA biosynthesis assessment, cell viability assay, and enzyme assay in microsomes can also be applied because researchers can obtain each reagent and instrument required for examination [102-105]. As extracellular AGEs may be able to promote ulcers in three kinds of epithelial cells via AGEs-RAGE/TLR4 axis, although this may also be caused by other factor (such as anti-cancer drug side effects [7]). Researcher may employ AGEs obtained from manufacturers or prepared by themselves as models of AGEs in saliva, blood, food, and beverage against cells [106-108], and the methods is applicable to extracellular as well as intracellular AGEs.”
(Lines 283-318)
We inserted the new references 2, 3, 68,69, 72, and 88-108 in this section.

Reviewer 2 Report
The manuscript ID metabolites-2506715 is entitled: “Novel in vitro Assay of the Effects of Kampo Medicines against Intra/Extracellular Advanced Glycation End-Products in Oral, Esophageal, and Gastric Epithelial Cells”.
The authors described the results of Kampo Medicines is not well understood yet. Their components are affected through digestion, absorption, and metabolism. However, in vitro oral, esophageal and gastric epithelial cell models avoid these steps, making them superior assay systems. The study focused on two strength points of this assay system against intracellular and extracellular advanced glycation end-products (AGEs).
The authors describe experiments to conclude the evolution of the methods and how they act in the formation of the final products in intracellular e extracellular. This manuscript is a good font of This manuscript is a great evolution for updating the AGEs.
In conclusion, the authors consider this subject as great development through figures 1 to 4 for justify how the components act in the different types of AGE absorption.
The manuscript is suitable for publication in Metabolites.
No comments
Author Response
Response Letter to Reviewers’ Comments
Responses to Reviewer 2
Dear Reviewer 2,
Thank you for giving us the opportunity to submit a revised draft of our manuscript titled “Novel in vitro Assay of the Effects of Kampo Medicines against Intra/Extracellular Advanced Glycation End-Products in Oral, Esophageal, and Gastric Epithelial Cells” to the journal Metabolites (manuscript ID: 2506715).
Comment: No comment
Response: Thank you for your evaluation of this manuscript against my Review article. Since we rewrote the manuscript based on other Reviewers’ comments, I believe that the revised manuscript will be accepted; I believe that the quality of the revised manuscript is significantly increased over that of the previous version. We inserted the new references 2, 3, 66-69, 72, and 88-108 in the revised manuscript.

Reviewer 3 Report
1. "Kampo medicine is a Japanese traditional medicine uniquely developed from Chinese traditional medicines."The word "uniquely" is unnecessary. The fact that Kampo medicine is derived from Chinese traditional medicines is already implied by the phrase "Japanese traditional medicine."
2. What new insights or advantages does it offer? Emphasize the unique contribution of this research to the existing literature.
3. Explain why the focus of the study is on methylglyoxal and glyceraldehyde-derived AGEs. What is their relevance to the overall research objective?
4. How does this contribute to our understanding of AGE-related diseases?
5. Review the text for grammatical errors, awkward phrasing, or ambiguous statements. Clear and concise writing will improve the overall quality of the manuscript.
Author Response
Response Letter to Reviewers’ Comments
Responses to Reviewer 3
Dear Reviewer 3,
Thank you for giving us the opportunity to submit a revised draft of our manuscript titled “Novel in vitro Assay of the Effects of Kampo Medicines against Intra/Extracellular Advanced Glycation End-Products in Oral, Esophageal, and Gastric Epithelial Cells” to the journal Metabolites (manuscript ID: 2506715). We appreciate the time and effort that the reviewers have dedicated to providing their valuable feedback on our manuscript. We are grateful to the reviewers for their thoughtful suggestions and insightful comments on our paper, which have enriched the manuscript and produced a better and more balanced account of the research.
We inserted the new references 2, 3, 66-69, 72, and 88-108 in the revised manuscript.
Comment 1: "Kampo medicine is a Japanese traditional medicine uniquely developed from Chinese traditional medicines. "The word "uniquely" is unnecessary. The fact that Kampo medicine is derived from Chinese traditional medicines is already implied by the phrase "Japanese traditional medicine."
Response 1: This has been revised accordingly in the Abstract and Introduction sections.
(Lines 12 and 31)
Comment 2: What new insights or advantages does it offer? Emphasize the unique contribution of this research to the existing literature.
Response 2: Following your suggestions, we have added the following sentences in Section 5. 3.“Potential of the novel assay”.
“This novel assay is suitable for the investigation of the relationships between lifestyle-related diseases or cellular dysfunctions, intracellular/extracellular AGEs, and the effects that Kampo medicines may exert against them. The effects of Kampo medicines can be assessed while bypassing the influence of digestion, absorption, and metabolism on the examined compounds. While other, unrelated natural products such as different Eastern traditional medicines may also be assessed in this manner, we consider that Kampo medicines represent the more suitable candidates due to their ensured quality and conformance to the Japanese Pharmacopoeia [2,3]. We suggest that the mechanisms of intracellular/extracellular AGEs have the potential to induce ulcers and to promote tumors in three kinds of epithelial cells. Although the cytotoxicity of intracellular AGEs in oral and esophageal cells remains unclear, clear relationships between such AGEs and gastric ulcers and tumors have been reported [35,88,89]. Naito et al. and Takagi et al. reported that argpyrimidine-modified peroxidoxin VI protein was detected in gastric cells in a diabetic model mice, and found that methylglyoxal modification of proteins delayed gastric ulcer healing [88,89]. Oya-Ito et al. detected argpyrimidine-modified HSP25 in RGK-1 cells, but not in RGM-1 cells [35]. Because these AGEs-modified proteins are of the argpyrimidine-modified type, we propose that rapidly generated AGEs may be suitable targets for such investigations into AGEs associations with lifestyle-related disease or organ dysfunction. Because this in vitro assay constitutes a simple system, the cells of which can be incubated in medium in dishes or plates, researchers can make use of it to treat the cells with metabolites/intermediate compounds from the glucose and fructose metabolism, and collect both cells and supernatant. Intracellular AGEs can then be identified and quantified by western blot [30,63], immunostaining [26,32], slot blot [68-70], ELISA [28,71-73], GC-MS [64], MALDI-MS [35,74], and ESI-MS [33,75-77]. mRNA levels, intracellular protein levels, secreted/released proteins (e.g. cytokine), and type of cell death (e.g. necrosis, apoptosis) can be determined with PCR [90,91], western blot [92,93], slot blot [68], ELISA [94-96], MALDI/ESI-MS [92], microscope [97,98], or flow cytometry [97, 99-101]. Specific investigation methods such as the measurement of membrane potential, DNA biosynthesis assessment, cell viability assay, and enzyme assay in microsomes can also be applied because researchers can obtain each reagent and instrument required for examination [102-105]. As extracellular AGEs may be able to promote ulcers in three kinds of epithelial cells via AGEs-RAGE/TLR4 axis, although this may also be caused by other factor (such as anti-cancer drug side effects [7]). Researcher may employ AGEs obtained from manufacturers or prepared by themselves as models of AGEs in saliva, blood, food, and beverage against cells [106-108], and the methods is applicable to extracellular as well as intracellular AGEs.”
(Lines 283-318)
We inserted the new references 2, 3, 68,69, 72, and 88-108 in this section.
Comment 3: Explain why the focus of the study is on methylglyoxal and glyceraldehyde-derived AGEs. What is their relevance to the overall research objective?
Response 3: Please note that we did not focus on only methylglyoxal and glyceraldehyde-derived AGEs (MGO-AGEs and GA-AGEs). Rather, in this article, we targeted AGEs which are suitable for analysis using the novel assay, which includes categorizations such as MGO-AGEs and GA-AGEs. We focused on individual AGEs based on their structure and origin, and on rapidly generated AGEs such as argpyrimidine-, MG-H1-, GLAP-, and CEL-modified proteins in cells (these AGEs can be generated from methylglyoxal and/or glyceraldehyde.) However, CML could also be rapidly generated from glycolaldehyde in MC3T3-E1 cells [76].
We deleted the words “methylglyoxal and glyceraldehyde-derived AGEs” in the Abstract and Introduction sections, and inserted “some AGEs” instead (Lines 20 and 73).
For further clarification, we inserted the following passage in section 4.2:
“Current technology does not allow the analysis of complete intracellular AGEs in cells, however more limited analysis of AGEs is possible under specific conditions. We focused on several AGEs that can be generated to persist for a short period (1–2 days) [21,33,76,78]; in comparison, the turn-over time of oral and gastric epithelial cells is 7–14 days [36,37]. Methylglyoxal, glyceraldehyde, and glycolaldehyde are able to rapidly generate AGEs [61,62,76,79].”
(Line 220-225)
“Suzuki et al. reported that glycolaldehyde was able to generate CML in MC3T3-E1 (murine osteoblast cell line) within 24 h, and that CML may be rapidly generated if cells quickly produce high amount of glycolaldehyde [76].”
(Line 229-231)
Comments 4: How does this contribute to our understanding of AGE-related diseases?
Response 4: Following your suggestions, we have added an expanded discussion in Section 5. 3.“Potential of the novel assay”. Please see Response 2.
Comment 5: Review the text for grammatical errors, awkward phrasing, or ambiguous statements. Clear and concise writing will improve the overall quality of the manuscript.
Response 5: The revised manuscript was proofread by a Scientific English editing company (Editage Co.).

Round 2
Reviewer 1 Report
I'm completely satisfied with the answers of the authors.
Reviewer 3 Report
It is ok